# Maternal Chronic Ethanol Exposure Decreases Stress Responses in Zebrafish Offspring

**DOI:** 10.3390/biom12081143

**Published:** 2022-08-19

**Authors:** Juliet E. Kitson, James Ord, Penelope J. Watt

**Affiliations:** 1School of Biosciences, University of Sheffield, Sheffield S10 2TN, UK; 2Centre for Fish and Wildlife Health, University of Bern, 3012 Bern, Switzerland

**Keywords:** alcohol, development, transgenerational, cortisol, zebrafish, stress

## Abstract

In humans, prenatal alcohol exposure can cause serious health issues in children, known collectively as Foetal Alcohol Spectrum Disorders (FASD). Despite the high prevalence of FASD and a lack of effective treatments, the underlying mechanisms causing the teratogenic action of ethanol are still obscure. The limitations of human studies necessitate the use of animal models for identifying the underlying processes, but few studies have investigated the effects of alcohol in the female germline. Here, we used the zebrafish *Danio rerio* to investigate the effects of chronic (repeated for seven days) exposure to alcohol. Specifically, we tested whether the offspring of females chronically exposed to ethanol during oogenesis exhibited hormonal abnormalities when subjected to a stressor (alarm cue) as larvae, and if they exhibited anxiety-like behaviours as adults. Exposure to alarm cue increased whole-body cortisol in control larvae but not in those of ethanol-treated females. Furthermore, adult offspring of ethanol-treated females showed some reduced anxiety-like behaviours. These findings suggest that the offspring of ethanol-treated females had reduced stress responses. This study is the first to investigate how maternal chronic ethanol exposure prior to fertilisation influences hormonal and behavioural effects in a non-rodent model.

## 1. Introduction

In humans, prenatal alcohol exposure has been directly linked to a number of behavioural, physical and cognitive birth defects, collectively known as Foetal Alcohol Spectrum Disorders (FASD) [1,2]. These alcohol-related deficits are of concern globally [3], have been estimated to affect 2–5% of children born in the USA [4] and are recognised as the leading non-genetic cause of learning difficulties in the UK [5]. These debilitating conditions are permanent, and available treatments are currently limited to early diagnosis and intervention strategies [6].

Despite the high prevalence of FASD and a lack of effective treatments, the mechanisms causing teratogenic sensitivity to ethanol are still relatively obscure [7]. This, coupled with the limitations of human case studies [8], has led to a surge in research utilising animal models to profile these vulnerable processes. Although there has been much research on rodents and the impact of prenatal exposure (e.g., reviewed in [7,9]), the zebrafish, *Danio rerio*, has been shown to be a useful model in which to investigate the mechanistic impacts of FASD. Zebrafish are easy to keep, eggs develop ex situ*,* so are not affected by any placental influence or parental care [10,11], and ethanol metabolising genes [12,13] as well as brain structure and function [14] are evolutionarily conserved between zebrafish and humans. Additionally, zebrafish complex behaviours can be used to screen for the characteristic effects of moderate developmental exposure to ethanol [15,16].

Ethanol treatment of embryos causes robust FASD-like defects [17,18,19]. Indeed, extreme embryonic exposure to ethanol produces similar craniofacial and skeletal deformities in zebrafish to those that are classically associated with human FASD (for review see [17]). Similarly, archetypal FASD behavioural deficits in the absence of physical deformities, such as reduced social (shoaling) and anxiety-like behaviours [20,21], are evident in zebrafish that were moderately exposed to ethanol during development. These behavioural deficits persist into later life [22,23] and endocrine responses to stressful stimuli, quantifiable using cortisol, the primary stress steroid in both humans and zebrafish [24,25], are reduced [26]. Additionally, behavioural and cognitive defects associated in rodents with prenatal ethanol exposure have been linked to epigenetic mechanisms [3] and have been found to be transmitted transgenerationally via the male germline to unexposed third-generation offspring [27,28]. Given this long-term persistence, capacity to alter epigenetic profiles and transmission to subsequent generations in rodents, it seems plausible that both behavioural and endocrine deficits associated with ethanol exposure could persist into the next generation in zebrafish; however, this remains to be fully explored. Furthermore, few studies have specifically looked at the transgenerational effects of alcohol via the female germline [29]. Recent studies have shown that parental dietary zinc deficiency [30], poor quality diet [31] and exposure to crude oil and lead toxins [32] all have the potential to induce behavioural and physiological deficits in zebrafish offspring. Indeed, Suresh et al. [33] exposed adult males and females to ethanol and detected changes in the behaviour of their offspring.

We therefore investigated whether chronic exposure to ethanol in adult female zebrafish induced behavioural and endocrine alterations in their offspring. We hypothesised that larval offspring of ethanol-exposed females would exhibit reduced cortisol levels in response to a stressful stimulus (a fear inducing alarm cue) compared with controls, as was found when embryos were treated directly with ethanol [26], and that anxiety-like behaviour would be affected in adult offspring. We focused on females to eliminate sex-specific differences [34,35] and because of the limited knowledge of ethanol via the maternal germline [29].

## 2. Materials and Methods

### 2.1. Study Species

Adult London Wild Type (LWT) females between 6–10 months old from stock housed at the University of Sheffield were used in the experiments. Stock LWT males of the same age were used for mating. Fish were kept in 10-litre tanks (30 cm × 15 cm × 24 cm) on a recirculation system in groups of 15 individuals. Water was maintained at 27 °C and lighting was on a 14:10 h light:dark (08:00–22:00 light:22:00–08:00 dark) photoperiod regime. Fish were fed twice daily on brine shrimp (*Artemia* sp.) and dry fish flakes.

### 2.2. Chronic Ethanol Exposure

Females were mated 24 h before exposures began to release any stored eggs and stimulate oocyte production. In total, 28 females were used in the experiments, i.e., 14 for the control and 14 for the ethanol-treated groups. An intermittent exposure approach was taken to more accurately mimic what human drinkers would experience [36], as has been used previously [37], as opposed to a continuous treatment. All control and treatment fish were gently netted into a covered exposure tank (22.5 cm × 14.5 cm × 14 cm) in groups of 7 containing 2 L of either aquarium water (control) or 1% ethanol (Fisher Scientific UK Ltd., Loughborough, UK) in aquarium water (ethanol-treated) for 30 min. Fish were then transferred to a tank containing aquarium water for 15 min to remove any traces of ethanol (in the case of the ethanol treatment) or to mimic the potential handling stress (in the case of the controls) before being returned to the housing tanks. By treating the controls in the same way as the treatment fish, any stresses that were incurred through handling and could have affected baseline behaviours were controlled for. The exposure procedure was repeated for 7 days and no physical abnormalities were observed following this. This repeated exposure to ethanol over a 7-day period is subsequently referred to as “chronic”. This procedure relies on the ethanol entering the tissues of the adult zebrafish and exposing the eggs indirectly via this route. Although little is known about how much ethanol circulates in the blood after exposure, significant levels have been reported in the brain after 15 min and levels may be 90% of that found in the surrounding water (Reviewed in [38]). Previous research has shown that when very early embryos (24 h PF) are exposed to 1% ethanol, although the concentration inside the egg was only 0.04%, the behaviour of the fish was affected [20], suggesting only small concentrations of ethanol are required to induce an effect. Behavioural tests (novel tank diving and open field) were conducted on the ethanol-treated and control females within 24 h of the final exposure to minimise any confounding effects of withdrawal [37]. Research was conducted under Home Office licence PPL 40/3704 held by P.J.W.

### 2.3. Breeding

Control and ethanol-treated females were placed individually into breeding tanks containing two untreated males per tank on the eighth day after their initial exposures so that eggs were produced on the morning of the ninth day. Forty-two eggs from each female were taken and distributed into two Petri dishes (21 eggs per dish) to be used for cortisol analyses of the larvae. The rest of the F1 generation were reared to adulthood in nursery tanks and female offspring (*n* = 14 ethanol, *n* = 7 control) were used in behavioural tests once they could be sexed at approximately six months of age.

### 2.4. Hormonal Responses of Larvae of Ethanol-Treated and Control Females to a Stressor

At 5 days post-fertilisation (DPF), larval offspring were exposed to a fear-inducing alarm pheromone extracted from the skin of euthanised fish [39,40,41,42], which is used to communicate the presence of danger to conspecifics, to determine cortisol reactions in response to a stressful stimulus. The alarm cue was extracted from epidermal cells of adult zebrafish using the protocol described by Jesuthasan and Mathuru [43]. Previous research has shown that exposure to alarm cue invokes a similar stress response to other stressors such as NaCl [44] and stirring [45]. Trials took place between 15:00 and 17:00. Two hundred μL of either aquarium water or alarm cue was pipetted into the centre of Petri dishes containing groups of larvae (from 16–21, depending on mortality of the original 21) and they were left for 15 min to allow for diffusion and to ensure all the larvae were exposed. In total, larvae were obtained from 10 control females and 10 ethanol-treated females. In all but one brood from the control females and two broods from the ethanol-treated females, which could not be divided in half because of low numbers, larvae were distributed between two Petri dishes and half were exposed to alarm cue and half to water so there were 19 dishes of larvae from control and 18 from ethanol-treated females. The one control brood that could not be halved (total brood *n* = 21) was exposed to water. For the two ethanol-treated female broods that had low numbers, one was exposed to alarm cue (total brood *n* = 20) and one was exposed to water (total brood *n* = 16).

Larvae were immediately sacrificed following the exposure to analyse cortisol levels in response to either water or alarm cue. Each group of larvae was immobilised using ice water, before being placed into 2-mL microfuge tubes and flash frozen in liquid nitrogen. Tubes were then stored at −80 °C (Forma Ultracold Freezer, Thermo Fisher Scientific, Asheville, NC, USA) to be used for whole-body cortisol extraction and analysis. Whole-body cortisol extraction and ELISA were carried out using the methods described by Yeh et al. [46]. Briefly, the groups of larvae stored at −80 °C were thawed and homogenised for 20 s using a mixing block (MB-102, Bioer, Hangzhou, China). Ethyl acetate (VWR International, Lutterworth, UK) was added to the homogenate, vortexed and centrifuged (Biofuge Pico, Heraeus, Hanau, Germany) at 10,000× *g* for 5 min. The supernatant was decanted and vaporised by incubating for 1 h at room temperature. Cortisol was then dissolved in 60 µL of 0.2% bovine serum albumin (Thermo Fisher Scientific, Loughborough, UK) in phosphate-buffered saline and stored at −20 °C. Cortisol ELISA assay was carried out the following day. Cortisol level for each group was adjusted to account for the number of larvae and recorded as pg per larva.

### 2.5. Behaviour of Ethanol-Treated and Control Females and Their Adult Female Offspring

Both adult females and their female offspring were tested for anxiety-like behaviours using the novel tank diving test and the open field test. Tests were carried out between 14:00 and 17:00. For the novel tank diving test, zebrafish were gently netted from their housing tanks and placed individually into a holding tank for three minutes to reduce handling stress [47]. Afterwards, fish were placed into a novel tank (23.5 cm × 8.0 cm × 23.0 cm) filled with 3 L of aquarium water for 5 min. Behaviour was immediately video-recorded using a Panasonic HC-X920 camcorder. Fish automatically swim to the bottom of a novel tank and so the time spent in the top half of the tank (seconds), and the time taken to first enter this area, defined as latency (seconds) [21,23,48], were quantified as a measure of anxiety, with reduced time in the top half and increased latency being indicative of anxious behaviour. Following the novel tank diving test, fish were immediately transferred into an open field experimental tank (22.5 cm × 14.5 cm × 14.0 cm, featuring a 4 × 5 grid on the base) filled with 2 L of aquarium water. Behaviour was video recorded for 5 min. Video recordings were analysed using automated tracking software (ViewPoint ZebraLab, Lyon, France). Heightened anxiety, recognised by reduced exploratory behaviour, was quantified by distance travelled (cm) using ViewPoint software. Time spent (s) and distance travelled (cm) in the outer zone, 3–4 cm from the edge, measured thigmotaxis, the extent to which individuals stayed close to the wall of the tank (an indication of anxiety). These parameters were selected based on previous studies of zebrafish behaviour [23,49]. Twenty-eight adult females (14 ethanol treated, 14 water treated controls) and 21 adult female offspring (14 from mothers exposed to ethanol (five different mothers), 7 from mothers exposed to water (three different mothers)) were tested.

### 2.6. Statistical Analysis

All data were analysed using R (R Foundation for Statistical Computing, Vienna, Austria, version 3.3.1) [50]. Offspring cortisol data were log-transformed due to positive skew and fitted to a mixed model with random effect of batch. Open field data were fitted to a linear mixed model with the fixed effect of ethanol treatment and either a single random effect of batch for parental data, or both batch and parent for offspring data, to account for any heritable variance. Parental time spent in the top half from the novel tank diving test was fitted to a zero-inflated Poisson generalised linear model to account for the overabundance of zero values and a number of high-value outliers in the positively skewed data distribution. Parental latency data from the novel tank diving were square-root transformed to address negative skew and fitted into a two-way ANOVA to analyse both the effect of ethanol exposure and of batch. Data from offspring novel tank diving tests were fitted into a generalised linear mixed Poisson model with random effects of batch and parent. The marginal *R*^2^ value was calculated using the R package MuMIn, and used to summarise the goodness-of-fit of the chosen mixed models [51].

## 3. Results

### 3.1. Chronic Ethanol Exposure

Female zebrafish exposed chronically to 1% ethanol did not spend significantly more time in the top half of a novel tank (Figure 1A) (estimate = 1.3, SE = 0.93, z = 1.4, *p* = 0.16) or show a significant decrease in latency to enter the top half of the tank (Figure 1B) (ANOVA: F = 3.78, df = 1,24, *p* = 0.06), which would have indicated reduced anxiety, compared with non-exposed controls. During the open field test, ethanol-treated females travelled, on average, 1320 cm more than controls (Figure 2A) (estimate = 1279.8, SE = 617, t = 2.081, df = 30.3, *p* < 0.05). However, chronic ethanol exposure did not significantly alter distance travelled in the outer zone (Figure 2B) (estimate = 14.8, SE = 9.9, t = 1.5, df = 27, *p* = 0.15) or time spent there during thigmotaxis analysis (Figure 2C) (estimate = 9.1, SE = 11.3, t = 0.8, df = 27, *p* = 0.43).

### 3.2. Hormone Responses of Larvae of Ethanol-Treated and Control Females to a Stressor

Offspring of control females exhibited significantly higher levels of whole-body cortisol when exposed to the alarm cue (df = 32, *p* < 0.05), as predicted (Figure 3). However, exposure of ethanol-treated mothers’ offspring to alarm cue had no significant effect on larval cortisol levels (df = 32, *p* = 0.57), and in fact they had an elevated cortisol level that was similar to the control alarm cue response. There was no interaction between maternal treatment and response to alarm substance (df = 32, *p* = 0.13). 

### 3.3. Behaviour of Ethanol-Treated and Control Females and Their Adult Female Offspring

The female offspring of ethanol-treated females spent, on average, ~41 s longer in the top half of a novel tank than the offspring of control females (Figure 4A) (estimate = 2.008, SE = 0.723, z = 2.780, *p* < 0.01). However, latency was not significantly affected by maternal treatment (Figure 4B) (estimate = −1.199, SE = 1.225, z = −0.978, *p* = 0.33*).* Maternal treatment did not significantly affect distance travelled (cm) by offspring in the open field test (Figure 5A) (estimate = −98.48, SE = 273.67, t = −0.360, df = 8.972, *p* = 0.73). Similarly, neither distance travelled in the outer zone (Figure 5B) nor time spent there (Figure 5C) were significantly affected by maternal exposure to ethanol (estimate = 7.918, SE = 14.62, t = 0.542, df = 8.382, *p* = 0.60 and estimate = 0.215, SE = 0.559, z = 0.384, *p* = 0.70, respectively).

## 4. Discussion

We investigated whether chronic exposure to ethanol in adult female zebrafish induced hormonal and behavioural alterations in their offspring. We show that exposure to alarm cue caused significant cortisol elevation in control larvae but not in those produced by ethanol-treated females, though the interaction between female treatment and offspring treatment was not statistically significant. Elevated whole-body cortisol levels have been previously linked to alarm cue exposure in adult zebrafish [24,52]. The apparent lack of a response in the ethanol-treated females’ offspring and their elevated baseline cortisol level would correspond with previous observations of impairment in danger perception and stress response in zebrafish following low acute alcohol exposure [26,53] and it is possible that they could be in a hyper stressed state. Alcohol exposure is known to cause dysregulation of the hypothalamic–pituitary–adrenal (HPA) axis, which controls the stress response in rodents [54,55]. Indeed, zebrafish with a disrupted hypothalamic–pituitary–interrenal (HPI) axis, the fish equivalent of the HPA, fail to respond to stressors both behaviourally and hormonally [44], suggesting that the stress response is affected in the animals exposed to ethanol. Though it is still unclear how the perception of danger is affected by ethanol exposure, it is conceivable that the mechanism could persist in the next generation and contribute to the behavioural patterns observed in this study. The mechanisms have not been definitively identified, though there is evidence of alcohol causing changes in the methylation of specific genes, including those associated with the stress axis, at least in the male germline [29].

The baseline cortisol levels in the offspring of the ethanol-treated fish were relatively high compared to the offspring of untreated females. Abnormal cortisol levels can lead to developmental defects in embryos [56] and impaired HPI axis functioning [57]. Following ethanol exposure, maternal cortisol could have been deposited in the ovarian follicles, though we did not measure this because it would have meant sacrificing the fish, but has been found in zebrafish fed with cortisol-spiked diets [56]. Maternal regulation of elevated cortisol levels in ovarian follicles is thought to be adaptive and to protect the embryos against the detrimental effects of excessive cortisol [56,58], but it is possible that chronic ethanol exposure disrupts this process, resulting in high baseline cortisol in offspring. Indeed, Nesan and Vijayan [57] have suggested that maternal stress and the consequent transfer of excess cortisol to the embryos could affect the HPI axis functioning in zebrafish larvae, possibly due to altered neurogenesis [59]. Previous research has shown that zebrafish larvae treated with cortisol have chronically elevated glucocorticoid signalling and maintain high levels of cortisol in adulthood leading to permanently and constitutively upregulated pro-inflammatory genes and a compromised immune and inflammatory response [60].

Adult female offspring of ethanol-treated females exhibited reduced anxiety during the novel tank diving test, as indicated by an increased allocation of time in the top half [48], but latency to enter the top half and other measures in the open field test were not affected by maternal treatment. This suggests that ethanol can induce reduced anxiety-like behaviours in the offspring of exposed individuals to some extent, but is contrary to work on rodents that has shown an increase in these behaviours [28] and an increased hormonal response [29] in offspring after female exposure. Tran et al. [61] suggest that chronic ethanol exposure in adult zebrafish attenuates dopamine, serotonin and associated metabolite reactions to secondary ethanol exposure. Similarly, developmental exposure to ethanol has been linked to shifts in baseline serotonin and dopamine [62,63], which in turn can influence social behaviour in zebrafish [64] and reduce their motivation to socialize [65]. It is therefore plausible that permanent shifts in baseline dopamine and serotonin resulting from ethanol exposure could persist in the next generation and influence some of the patterns observed in this study. Furthermore, excess cortisol transferred to the embryos following maternal stress may lead to a reduced stress response [66], an increase in boldness [59] and a compromised immuno-inflammatory response [60].

The anxiolytic action of ethanol following chronic exposure was only observable in parental zebrafish females during the open field test. Ethanol-treated females travelled a greater distance, indicative of exploratory behaviour and reduced anxiety, corroborating previous findings [23]. Several studies have found evidence of behavioural adaptation to ethanol following prolonged chronic exposure [61,67,68,69], and so anxiolytic effects may not have been observable in all tests for this reason.

This study is the first to investigate how maternal chronic ethanol exposure prior to fertilisation influences hormonal and behavioural effects in zebrafish offspring. We found evidence for an altered stress response and changes in some anxiety-like behaviours in the offspring of ethanol-treated females, suggesting that the effects of alcohol can be transmitted across generations in a fish. This study illuminates an approach to investigating the action of ethanol that has not previously been explored, and which may prove valuable in further elucidating the teratogenic risks associated with ethanol consumption.

## Figures and Tables

**Figure 1 biomolecules-12-01143-f001:**
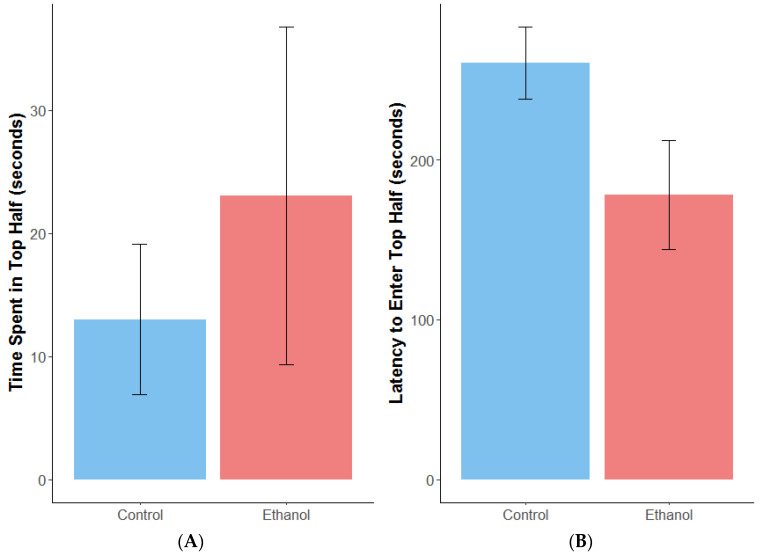
Mean (+/−SE) allocation of time spent in the top half of the tank (**A**) and latency to enter the top half of the tank (**B**) for adult female zebrafish during the novel tank diving test after exposure to 1% ethanol or water (control). *n* = 14 for the control and 14 for the ethanol-treated females.

**Figure 2 biomolecules-12-01143-f002:**
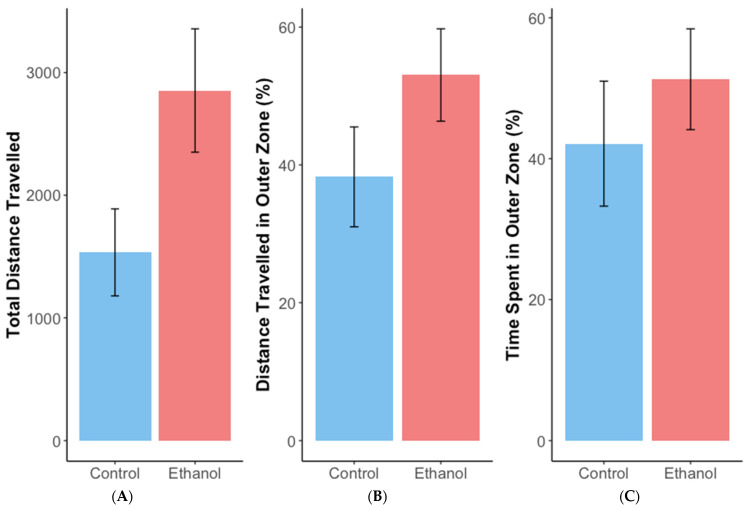
Mean (+/−SE) total distance travelled (**A**), and distance travelled (**B**) and time spent (**C**) in the outer zone as a percentage of the total for adult female zebrafish during the open field test after exposure to 1% ethanol or water (control). *n* = 14 for the control and 14 for the ethanol-treated females.

**Figure 3 biomolecules-12-01143-f003:**
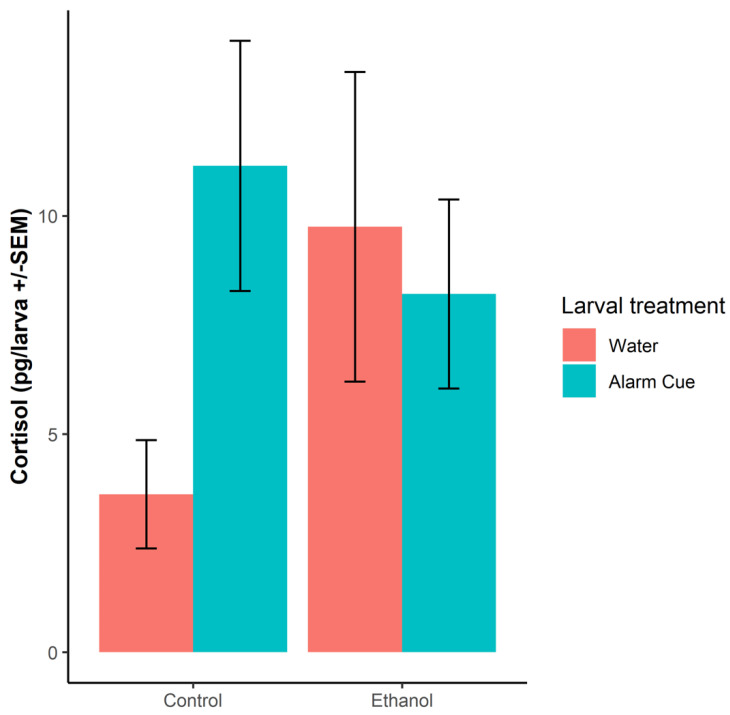
Mean (+/−SE) cortisol levels of larvae from either control or ethanol-treated mothers following exposure to either water or alarm cue, respectively. In total there were 19 dishes of larvae from control and 18 from ethanol-treated females. Dishes were exposed to alarm cue or water. Each dish contained between 16–21 larvae.

**Figure 4 biomolecules-12-01143-f004:**
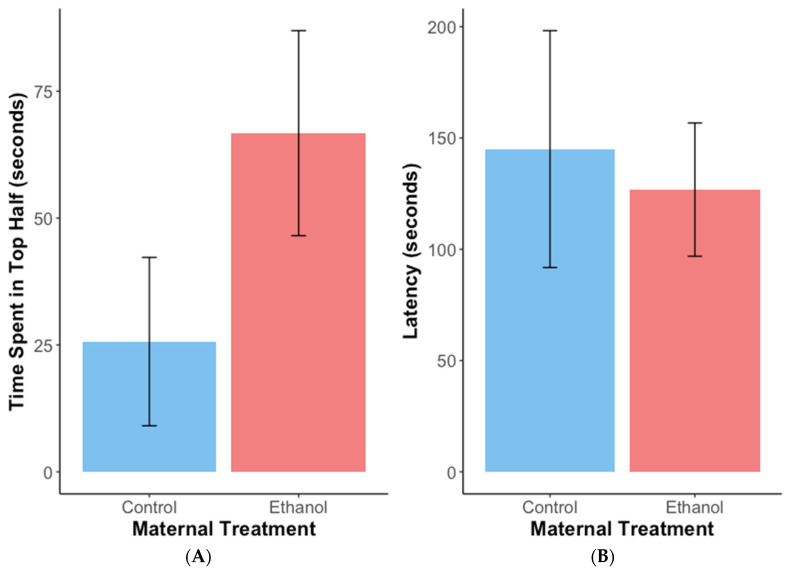
Mean (+/−SE) time spent in the top half of a novel tank (**A**) and latency to enter this half (**B**) for adult female offspring from mothers exposed to ethanol (*n* = 14) or water (control) (*n* = 7) during the novel tank diving test.

**Figure 5 biomolecules-12-01143-f005:**
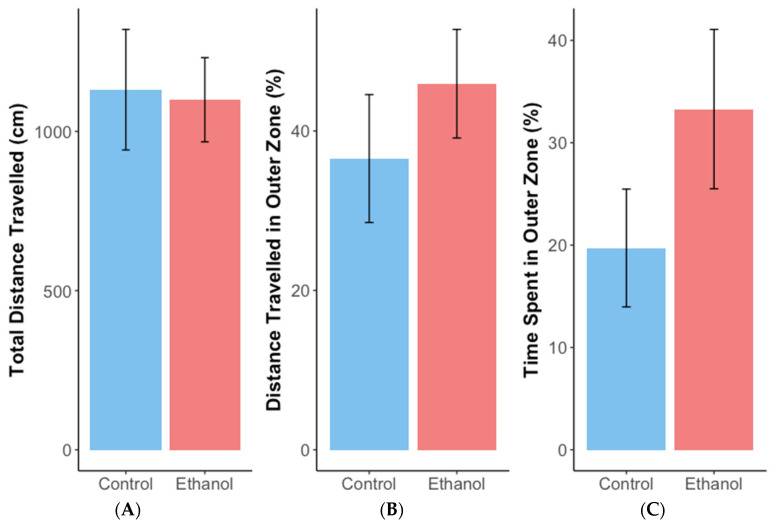
Mean (+/−SE) distance travelled (**A**), and distance travelled (**B**) and time spent (**C**) in the outer zone as a percentage of the totals for adult female offspring from mothers exposed to ethanol (*n* = 14) or water (control) (*n* = 7) during the open field test.

## Data Availability

Data is stored on ORDA, the University of Sheffield’s data repository.

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
