# Peer review of "Maternal Chronic Ethanol Exposure Decreases Stress Responses in Zebrafish Offspring"

_biomolecules, 2022, doi:10.3390/biom12081143_

Round 1

Reviewer 1 Report

In this paper, authors report chronic exposure of ethanol in female zebrafish caused the offspring larvae to become more explorative, less feared with reduced level of cortisol response against a stressor. The effect is clearly demonstrated and intriguing. However the underlying mechanisms are not clearly demonstrated. Firstly, why did author exposed alcohol to mother but not to father. In case of human FADS, pregnant mother drink alcohol that directly affect the developing embryos and foetus. So, it is a developmental exposure of ethanol to the embryo. But in the adult female zebrafish, eggs were possibly exposed to the alcohol by the ethanol treatment in water but embryos were not. Therefore the effect is assumed being an indirect consequence of epigenetic modification or the alteration in the egg composition. If it is epigenetics, it has been known that paternal epigenetic signature is more likely to affect the offspring than the maternal counterpart. Therefore it may be more sensible (or at least worth testing) in the paternal exposure too. If authors consider cortisol level is regulated by alcohol in the offspring larvae, what is the mechanism by which the level is upregulated? Author should provide some evidence about the alteration of the regulatory mechanisms of the cortisol (e.g. DNA methylation of genes involved in the cortisol metabolism)  

Reviewer 2 Report

In their manuscript “Maternal chronic ethanol exposure decreases stress responses in zebrafish offspring,” Kitson et al., show that chronic ethanol exposure in adult female zebrafish results in increased basal cortisol levels and decrease stress responses in female offspring of the exposed adults. Adult females were exposed to 1% ethanol intermittently in their tank water for 30 minutes a day for 7 days. These females (along with controls) were allowed to mate and embryos from these crosses were tested for hormonal stress levels. In addition, the ethanol exposed adults and the adult offspring of the ethanol exposed adults showed maladaptive behaviors in regards to stress response behaviors. The authors’ results will be of interest to the readership of Biomolecules, and the broader ethanol birth defects research community. However, several concerns exclude publication of the manuscript in its current form.

Major Concerns:

1.    The authors state that cortisol levels in the larval offspring of ethanol exposed adults does not change in response to the alarm cue. However, the basal level of cortisol is comparably increased compared to alarm cue control larva. This suggests that basal cortisol levels are increased in larva from exposed adults. The authors suggest this is due to high variability in the data. The authors need to show the extent of the variability in their data and how this explanation is the strongest, instead of increased cortisol levels in larva of ethanol exposed adults is simply being the new basal level of cortisol. Also is this consistent as the larva continue to develop or could this be an increased response based on maternal deposition or due to actual changes in the larva themselves?

2.    The authors should also should expand the discussion on chronic stress levels due to increased cortisol and how that may impact other hormonal responses, like dopamine and serotonin.

3.    The FASD research community is actively moving away from any language that may be stigmatizing to mothers regarding drinking alcohol while pregnant. The current approach to these events is to remove the mother altogether. The authors need to amend their language to describe FASD as “prenatal alcohol exposure” or similar language not describing the consumption of ethanol by the mother.

Minor concerns:

1.    Line 123: Please include 12 hr clock readings with the 24 hr clock readings

2.    Line 137: at what temp was the supernatant incubated

3.    Line 190: Please define thigmotaxis for any non-behaviorist reader

Reviewer 3 Report

The authors used a zebrafish model to investigate whether chronic exposure to ethanol in females causes behavioral and endocrine changes in their offspring. The ethanol-treated larvae showed higher alarm tolerance and reduced anxiety compared to the non-treated females. An increase in the cortisol level characteristic of stress was not observed in such fish. The main results of the study suggest that the effects of alcohol on fish can be passed on to offspring. The authors believe that alcohol causes dysregulation of the hypothalamic-pituitary-adrenal system, which leads to a decrease in the response to stress. They note that their study is the first to investigate how chronic ethanol exposure prior to fertilization affects hormonal and behavioral responses of zebrafish offspring, suggesting their experimental model could be helpful in further studies of human birth defects, collectively known as FASDs.

Potentially interesting, this paper therefore raises some questions that I would like to address to the authors.

1. I believe that beyond the notes in lines 41–45, the authors could better explain why the fish model is useful to study human FASDs if there is a perfect model of rats that actually consume ethanol? But fish swim in ethanol solution — does it get into the body? It is then necessary to take a test for the alcohol content in the fish.

2. From my point of view, the observed changes may be associated not only with the specific effect of alcohol, but also with a nonspecific reaction to stress, which, in fact, is based on a change in the activity of the adrenal cortex or the steroidogenic tissue in the kidney area of fish. As a control, one could take any stressor, such as a change in temperature, and see if the effect is different.

3. An important and complex issue concerns the transmission of behavioral responses from the female to offspring. In my opinion, the reactions are most likely linked to hormone levels and therefore are secondary. In mammals, the hormonal state of the female can definitely affect the hormonal level of the offspring. It is worth further explaining how this occurs in fish. By the way, the authors determined the cortisol level in eggs (Materials and Methods, lines 110–112), but I did not see these data in the text, at least in the form of histograms. An alternative interpretation of the authors’ observations can be proposed: the offspring do not actually become resistant to stress, they are simply in a state close to distress; the level of cortisol is chronically elevated, the adrenal cortex ‘runs ragged’, so the fish can no longer fully respond to stress.

4. It is not entirely clear from the text how the offspring came from different females were averaged. The experiments included 14 females for the control and 14 for the ethanol-treated groups. Did the authors get larvae from all these fish? The experiments used n = 15 ethanol, n = 6 control. The question remains open: the offspring of how many females were analyzed?

Round 2

Reviewer 2 Report

The authors have addressed all concerns. the mnauscript is ready to publish in current form.

Author Response

Thanks

Reviewer 3 Report

The authors have answered my questions and made appropriate additions to the text of the article, which made it clearer and scientifically sound. Therefore, I believe the MS has been sufficiently improved to be published in Biomolecules.

Author Response

Thanks